# The Influence of Histologic Inflammation on the Improvement of Liver Stiffness Values Over 1 and 3 Years

**DOI:** 10.3390/jcm8122065

**Published:** 2019-11-24

**Authors:** Jeong-Ju Yoo, Yeon Seok Seo, Young Seok Kim, Soung Won Jeong, Jae Young Jang, Sang Jun Suh, Hyung Joon Yim, Ki Tae Suk, Dong Joon Kim, Kwang-Hyub Han, Seung Up Kim, Bora Lee, Sang Gyune Kim

**Affiliations:** 1Division of Gastroenterology and Hepatology, Department of Internal Medicine, Soonchunhyang University Bucheon Hospital, Bucheon 14854, Korea; puby17@naver.com (J.-J.Y.); liverkys@schmc.ac.kr (Y.S.K.); 2Division of Gastroenterology and Hepatology, Department of Internal Medicine, Korea University Anam Hospital, Korea University College of Medicine, Seoul 02841, Korea; drseo@korea.ac.kr; 3Division of Gastroenterology and Hepatology, Department of Internal Medicine, Soonchunhyang University Seoul Hospital, Seoul 04401, Korea; jeongsw@schmc.ac.kr (S.W.J.); jyjang@schmc.ac.kr (J.Y.J.); 4Division of Gastroenterology and Hepatology, Department of Internal Medicine, Korea University Ansan Hospital, Korea University College of Medicine, Ansan 15459, Korea; mothpickle@naver.com (S.J.S.); gudwns21@korea.ac.kr (H.J.Y.); 5Department of Internal Medicine, Chuncheon Sacred Heart Hospital, Hallym University College of Medicine, Chuncheon 24253, Korea; ktsuk@hallym.ac.kr (K.T.S.); djkim@hallym.ac.kr (D.J.K.); 6Division of Gastroenterology, Department of Internal Medicine, Severance Hospital, Yonsei University College of Medicine, Seoul 03722, Korea; gihankhys@yuhs.ac (K.-H.H.); ksukorea@yuhs.ac (S.U.K.); 7Department of Biostatistics, Graduate School of Chung-Ang University, Seoul 06974, Korea; mintbora0125@gmail.com

**Keywords:** transient elastography, inflammation, liver biopsy

## Abstract

Background: Transient elastography is now an indispensable tool for estimating liver fibrosis. Although many clinical factors other than fibrosis itself are known to affect liver stiffness (LS) values, it is still not yet clear what factors are related to improving LS values. The aim of this study was to find out how baseline histologic inflammation influences LS values and how much this inflammation affects improvement in LS values over time, regardless of actual fibrosis content. Methods: This retrospective study included 678 consecutive patients who underwent liver biopsy and sequential LS assessment from 2006 to 2015 at six tertiary hospitals in Korea. Linear regression analysis was used to evaluate how improvement of LS value can be associated with other factors besides fibrosis content. Results: Basal LS values increased with increasing inflammation in the same fibrosis stage. Degree of inflammation influenced the baseline LS value in a proportional manner (beta coefficient (BE), 6.476; 95% confidence interval (CI), 2.24–10.72; *p* = 0.003). Moreover, histologic inflammation affected the change in LS value significantly. Higher inflammation grade at baseline was a significant predictor for an improvement in LS value, regardless of the fibrosis stage (BE, −8.581; 95% CI, −15.715–−1.447; *p* = 0.019). In a subgroup analysis of patients who received repeated liver biopsies, the results showed a similar tendency. Conclusions: The LS value is affected by the degree of inflammation even at a low ALT level. Furthermore, baseline histologic inflammation has a significant impact on the improvement of LS values over time. Therefore, baseline inflammation should be taken into consideration when interpreting an improvement in LS value.

## 1. Introduction

Transient elastography (TE; FibroScan^®^, Echosens, France) is the most commonly used noninvasive method for the assessment of liver fibrosis. However, the value of liver stiffness (LS) measured by TE is influenced by many factors other than fibrosis content. Typical confounding factors are hepatitis, mechanical cholestasis, liver congestion, cellular infiltration, and fat deposition. Among these, inflammation may increase the LS value through hepatocellular swelling and hepatic infiltration of the inflammatory cell [1,2]. Based on LS values continuously measured in the same patient, it was found that the lower the alanine aminotransferase (ALT) value, the lower the LS value. Likewise, since LS measurements (LSMs) by TE under hepatic inflammation are prone to overestimation, accurate assessment of the amount of intrahepatic fibrosis is hindered under these circumstances. In a previous study, LS value was estimated to increase by about 4 kilopascals (kPa) when ALT levels increased by about 100 U/L. That study suggested that valid assessment of fibrosis stage cannot be made by TE when ALT is greater than 100 U/L [2]. In another study, necroinflammation was found to be the main cause of discrepancies between liver biopsy results and LS values [3]. However, ALT is not the only factor that points to biochemical inflammation. In addition, ALT level does not always accurately reflect the severity of hepatitis. Previous studies have also shown that antiviral use, duration of therapy, higher initial LSM value, and hepatitis B virus (HBV) DNA levels are associated with improvements in measured LS values for patients with chronic hepatitis B (CHB) [4,5,6]. It is well known that inflammation affects LS values, but little research has been undertaken on how much histological inflammation contributes to the improvement of LS values. Several studies have shown that increased liver enzyme levels or elevated intrahepatic inflammation contribute to an increase of LSMs [7,8]. However, there is very little research showing that the degree of inflammation found in a liver biopsy actually affects the LS value and how changes in that LS value after long-term follow-up are closely related with the initial degree of inflammation. Our hypothesis was that LS is influenced not only by liver fibrosis, but also by inflammation and that patients with higher grade inflammation during baseline measurements would show a greater improvement in LS values regardless of actual fibrosis content. The aim of this study was to determine whether baseline hepatic inflammation affects LS measurements, independent of fibrosis stage, and identify factors associated with improvement in LS value over time.

## 2. Methods

### 2.1. Patients and Study Protocol

Between January 2006 and August 2015, we collected data from patients with chronic liver disease during routine clinical care at six tertiary hospitals. These data were taken from the electronic database of the Korean Transient Elastography Study Group. Patients who fulfilled the following inclusion criteria were eligible for this study: (a) patients who were clinically, pathologically, or radiologically diagnosed with chronic liver disease or liver cirrhosis; (b) patients who underwent an initial baseline liver biopsy; and (c) patients with repeated measurement of LS, including a baseline LS value taken at the time of their initial liver biopsy. Patients who met the following conditions were excluded from the study: (a) those who failed LSM; (b) those with aspartate aminotransferase (AST) or ALT levels of more than 5 times the upper limit of normal; (c) those who had inadequate liver biopsy samples; (d) those whose baseline liver biopsy data and baseline LSMs were more than 3 months apart; (e) those found to have hepatocellular carcinoma at enrollment or during follow-up; (f) patients whose repeated TE measurement was performed less than one year from their baseline LSM; or (g) patients who received antiviral therapy for hepatitis B or hepatitis C for less than one year. Finally, we retrospectively included 678 patients who met the criteria. The clinical, histological, and laboratory records of these patients were retrospectively reviewed.

This study’s protocol was approved by the institutional review board of SoonChunHyang University Bucheon Hospital (IRB number SCHBC 2018-01-001-001). The study protocol conformed to ethical guidelines set by the World Medical Association Declaration of Helsinki.

### 2.2. Liver Biopsy and Histology

Liver biopsy was performed when each patient’s doctor deemed it necessary to determine the cause and severity of their liver disease. Specimens were fixed in formalin and embedded in paraffin. Sections were stained with hematoxylin–eosin and Masson’s trichrome. Each biopsy specimen was analyzed by an experienced pathologist at each hospital. All pathologists were unaware of the TE results. Fibrosis was assessed to be at a stage from 0 to 4 on a scale using the METAVIR criteria where: F0, no fibrosis; F1, portal fibrosis without septa; F2, periportal fibrosis; F3, septal fibrosis; F4, liver cirrhosis. Histologic inflammation was graded using a modified histological index (HAI) [9,10]; none or minimal inflammation (inflammation score 0–4), mild (inflammation score 5–8), moderate (inflammation score 9–12), and severe (inflammation score 13–18). Specimens of at least 20 mm in length were considered eligible for interpretation in this study [11].

### 2.3. Transient Elastography

LS was measured by Fibroscan^®^ using M probe as reported previously [12]. Forty-two (6.2%) patients who did not qualify for M probe were measured by XL probe. The success rate was calculated as the number of valid measurements divided by the total number of measurements. Ten measurements were performed with a success rate of at least 60%. Results are expressed as kPa. The median value was taken as the representative value. Interquartile range (IQR) was defined as the index of intrinsic variability in the LS values and corresponds to the interval between the 25th and 75th percentiles, which contains 50% of the valid LSMs taken. Only procedures with at least ten valid measurements and a IQR/median value <0.3 were considered. LSMs were performed by expert physicians who had experience conducting these tests on more than 1000 cases. Improvement of LS value was defined as decrease of LS value compared to the previous ones.

### 2.4. Statistical Analysis

Frequencies and percentages were used for descriptive statistics. Statistical differences between groups were investigated using the χ^2^ test and Student’s *t*-test. Spearman’s analysis was used to investigate correlations between variables. To identify predictive factors associated with 1-year improvement in TE, logistic regression analysis was used. Multivariate models were created using variables that were significant (*p* < 0.10) in univariate analysis and clinically relevant. All statistical analyses were performed using R (version 3.3.3, The R Foundation for Statistical Computing, Vienna, Austria) and SPSS software (version 21.0; SPSS Inc., Chicago, IL, USA). Statistical significance was defined at *p* < 0.05.

## 3. Results

### 3.1. Baseline Characteristics

Baseline demographics and clinical characteristics of patients are summarized in Table 1. A total of 678 patients were analyzed, 329 (48.5%) were male and 349 (51.5%) were female. Mean age of these patients was 47.12 ± 12.25 years, with mean body mass index (BMI) of 23.97 ± 3.44 kg/m^2^. The most common cause of liver disease was HBV (61.8%), followed by hepatitis C virus (HCV) (27.0%), and then others (11.2%). Mean AST and ALT levels were 44 U/L and 46 U/L, respectively.

Distributions for fibrosis stage, steatosis, and inflammation grade in the study population are also presented in Table 1. The median LS value was 10.50 kPa. Distributions of LS value according to fibrosis stage, inflammation grade, and steatosis grade are shown in Figure 1. Baseline LS values were significantly correlated with fibrosis stage (*p* < 0.001 for trend) and inflammation grade (*p* = 0.001 for trend), but not with steatosis grade (*p* = 0.101 for trend) (Figure 1). Next, we investigated the distribution of LS value according to inflammation grade for patients in the same fibrosis stage (Appendix A). LS values were higher with higher inflammation grades within a same fibrosis stage (Appendix A).

### 3.2. Related Factors Determining Baseline LS Value

Based on biopsy results and laboratory findings, factors affecting baseline LS values were examined using linear regression analyses (Table 2). It was found that LS values increased with increasing fibrosis (beta coefficient (BE), 8.306; 95% confidence interval (CI), 2.730–13.882; *p* = 0.004). Furthermore, the degree of inflammation increased the LS value in a dose-dependent manner, regardless of fibrosis stage, total bilirubin, albumin, prothrombin time, or serum sodium level (BE, 6.476; 95% CI, 2.236–10.716; *p* = 0.003). It was also found that LS values tend to be lower in viral etiology than that in nonviral etiology for patients in the same fibrosis stage (BE, −3.56; 95% CI, −5.994 to −1.125; *p* = 0.004).

Since previous studies have shown that high ALT affects LS values, we separately analyzed patients with ALT values below 40 U/L. In 283 patients with normal ALT, fibrosis stage was well correlated with baseline LS value. In this group, histologic inflammation of greater than a moderate grade was also well correlated with LS value (Appendix A).

### 3.3. Factors Associated with Improvement of LS Value

We then analyzed factors associated with LS improvement in patients who had received antiviral agents consistently along with follow-up LSMs. We were able to obtain a follow-up LS value after one year from 358 patients and three-year follow-up data from 244 patients. The rate of change in the LS value was linear over one and three years (Figure 2). We investigated the change in LS value according to inflammation grade after one and three years (Figure 3). The LS value tended to improve significantly in relation to the degree of basal inflammation found at one-year and three-year follow-ups (both *p* < 0.001). Next, we performed linear regression analysis to find out which factors determine the change in LS value. The results of the linear regression analysis after one and three years are shown in Table 3 and Table 4, respectively. In our multivariate analysis, the higher inflammation group showed greater improvement in LS value at both one-year (BE, −8.581; 95% CI, −15.715–−1.447; *p* = 0.019) and three-year follow-ups (BE, −10.725; 95% CI, −19.299–−2.151; *p* = 0.014) after adjusting for other factors. In addition to inflammation, the factors related to changes in LS value were platelet levels (BE, 0.027; 95% CI, 0.011–0.042; *p* = 0.001), total bilirubin (BE, −1.716; 95% CI, −2.420–−1.011; *p* < 0.001), and prothrombin time (BE, −4.647; 95% CI, −6.126–−3.167; *p* < 0.001). Sensitivity analysis showed similar results with more significance (Appendix A).

### 3.4. Paired Liver Biopsy with Paired LSM

To accurately determine the effect of histological inflammation on LS values, we analyzed a subgroup of patients with paired liver biopsy and paired LS results (Figure 4). Follow-up biopsies were performed after a median period of two years (range one to eight years). Among the enrolled patients, there were 24 patients who had LSMs taken simultaneously with their repeated liver biopsies. A description of these 24 patients is presented in Appendix A. Among them, 12 patients showed improved LS values and 12 did not. Among the group with improved LS, 33.3% showed an improvement in fibrosis stage and 41.6% showed an improvement in inflammation grade on their follow-up liver biopsy (Figure 4). However, in the group without an improvement in their LS, histologic fibrosis and inflammation was improved in only 16.7% and 25.0% of patients, respectively (Figure 4). Although the group with decreased LS showed a higher proportion of improvement of histological fibrosis (33.3% vs. 16.7%) or inflammation (41.6% vs. 25.0%), these results were not statistically significant (fibrosis, *p* = 0.640; inflammation, *p* = 0.667). 

## 4. Discussion

TE was originally designed to accurately assess the amount of intrahepatic fibrosis without the need for invasive procedures. This procedure has now largely replaced biopsy for this purpose [13,14,15]. However, even if CHB patients with the same fibrosis stage and LS value are treated with the same antiviral drug, their improvements are different. Our research started with the question of why this difference would occur, despite the patients having the same fibrosis stage at baseline. Through this study, we found that LS values are affected even at low ALT level (<40 IU/L). Therefore, we need to pay attention to how we interpret LS values.

Many studies have reported that hepatic inflammation affects LS value, regardless of etiology [1,11,16,17]. However, previous reports were mainly focused on the association of LS values and ALT as an indicator of inflammation [7,8,18]. It is difficult to routinely perform liver biopsies except in patients with a specific indication. In this context, this study has important implications for evaluating whether a patient’s LS value is associated with their histologic inflammation grade rather than their ALT level.

As mentioned in the results, the degree of histologic inflammation is correlated with the improvement in LS value, but not with ALT level. This may be due to the fact that ALT level is not directly correlated with histologic inflammation in liver disease. This phenomenon is well-known in the immune tolerance phase of HBV. It has also been reported in other etiologies [19]. In HCV and nonalcoholic fatty liver disease, ALT is not a reliable marker for reflecting the degree of inflammation [20,21,22,23]. In the same vein, we should note that histologic inflammation in autoimmune hepatitis lasts for a considerable period of time even after normalization of ALT by biochemical remission [24]. We also found that baseline histologic inflammation was an important confounding factor affecting both the baseline LSM and subsequent improvement in LSMs, regardless of etiology.

There are several hypotheses explaining the discrepancy between histologic inflammation and ALT level. The first hypothesis is that ALT elevation is related to the location of inflammation [25]. When lobular inflammation is predominant, ALT levels will rise due to hepatocellular injury. However, if inflammation occurs mainly in the periportal area, any ALT rise will not be so noticeable. The second hypothesis is that changes in ALT level seem to be faster than histologic changes, especially in cases of autoimmune hepatitis, where biochemical remission has been reported to progress several months earlier than histological remission [26]. Considering that inflammation is the main mechanism of autoimmune hepatitis, the same phenomenon may apply to other etiologies where inflammation plays an important role in the pathophysiology. Third, the location of fibrosis, especially perisinusoidal fibrosis, can matter. Fraquelli et al. proposed this very hypothesis, their reasoning was that perisinusoidal fibrosis is not reflected in the METAVIR system and thus makes the interpretation of this histological confounder even more difficult [27].

In previous reports, it was found that inflammation affects LS values at low fibrosis stages but not at high fibrosis stages [3,28,29]. However, our results show that inflammation increases the LS value regardless of fibrosis stage, even in the cirrhotic stage. Therefore, if it is indicated that a patient may have cirrhosis by an LS value over 12 kPa, it is necessary to repeat the LSM after a little more time to allow any inflammation to settle down.

There are a few studies similar in design to our research. First, Verveer et al. [30] found that inflammation may result in high LS values, especially for patients in F1 or F2 stage fibrosis. However, this study differed from ours in that only patients with HBV and HCV were included. In addition, patients with a fivefold increase or more in ALT were included and this factor was not mentioned in terms of the change in LS value. Second, Liang et al. [31] found that a decline of liver stiffness is related with both liver inflammation and fibrosis. Paired biopsy results were also reported for 30% of patients. However, only patients with chronic hepatitis B were included and the follow-up period was set to two years, which was shorter than in our study. Finally, the study of Fraquelli et al. [27], mentioned above, was similar to our study in that necroinflammatory activity was seen to be a main determinant of TE. However, they did not mention any correlation between histologic findings and a change of LS value.

This study adds some knowledge that is not well understood in previous studies. First, it is well-known that LS values can be overestimated by inflammation if the ALT level is higher than about 200 IU/L in current clinical practice [7,8,18,32]. However, as seen in this study, patients with low ALT may have some degree of histological inflammation, and LS values may not be accurate in this situation. Second, baseline inflammation has a significant effect on longitudinal change in LS value. In patients with the same baseline LS value, patients with LS improvement over time are likely to have more severe initial histologic inflammation than those without improvement. In other words, the degree of inflammation has a significant effect on the LS value, regardless of the ALT level, and also has a significant effect on the change of the LS value.

In the case of F4 fibrosis, it was significantly correlated with baseline LS value as a snapshot, but it was not associated with how much LS value would be decreased or increased over time. This could be explained by two reasons. First, our study included heterogeneous etiology in addition to HBV and HCV. Second, in patients with antiviral therapy, we only included patients who received treatment for more than one year. The phenomenon of decreased LS value in patients taking antiviral treatment is mainly reported in early phase of antiviral treatment [33]. Therefore, the patients included in our study were likely to have relatively less improvement in TE values. A similar observation was also found with steatosis, where it does not influence the baseline LS value or change of LS value at one year, but severe steatosis does influence liver stiffness at three-year follow-up. This may be due to two reasons. First, severe steatosis has been reported to overestimate LS values [34,35], whereas mild to moderate steatosis has no significant effect on LSM value [36]. Second, it can be presumed that weight change has occurred more greatly in patients with severe steatosis over the course of three years. Unfortunately, we could not get enough information about weight change to analyze this association on account of it being a retrospective study.

There are some caveats to consider while interpreting our results. First, although we collected data prospectively according to protocol, they were analyzed retrospectively. Thus, selection bias is inevitable because of the lack of follow-up results in many patients. On the other hand, the strength of this study is that we got baseline histologic information and repeated liver stiffness measurement from all participants. Second, we did not perform paired liver biopsies on all patients. However, we presented repeated liver biopsy results together with corresponding LS values in a subgroup analysis. Although these results were not statistically significant due to the small sample size, the impact of inflammation on change in LS value could be supported. Third, this is a multicenter study in which several different pathologists made the pathologic readings, so we cannot confirm agreement between them. Finally, we did not evaluate alcohol consumption or weight change during follow-up testing, which could be confounding variables.

In summary, although LS is a relatively accurate method for assessing liver fibrosis in chronic liver disease, histologic hepatic inflammation has a significant impact on baseline measurement of LS values and their improvement. Therefore, initial LS values should be interpreted with caution since they are affected not only by fibrosis, but also by inflammation, even if it is at low degree of ALT level (<40 IU/L).

## Figures and Tables

**Figure 1 jcm-08-02065-f001:**
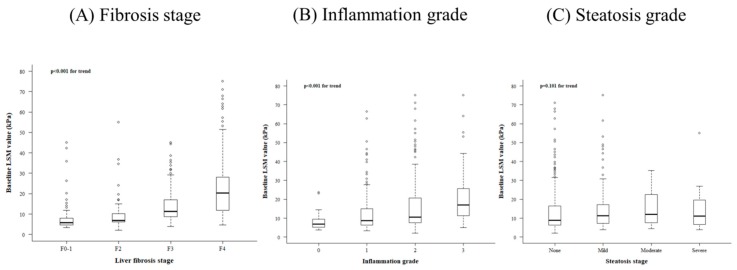
Distribution of liver stiffness (LS) values according to histologic findings. Distributions of LS values according to (**A**) fibrosis stage, (**B**) inflammation grade, and (**C**) steatosis stage are described in this figure. The length of the box represents the interquartile range, within which 50% of the value is located. The line through the middle of each box represents the median. Error bars show the minimum and maximum values (range).

**Figure 2 jcm-08-02065-f002:**
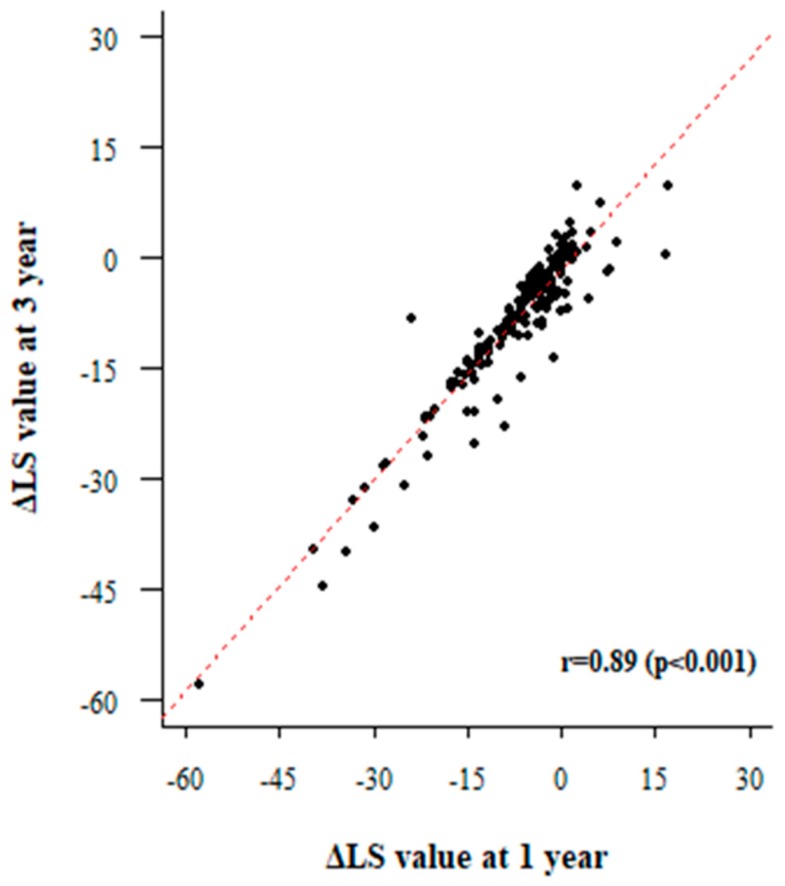
Correlation between 1-year liver stiffness value change and 3-year LS value change.

**Figure 3 jcm-08-02065-f003:**
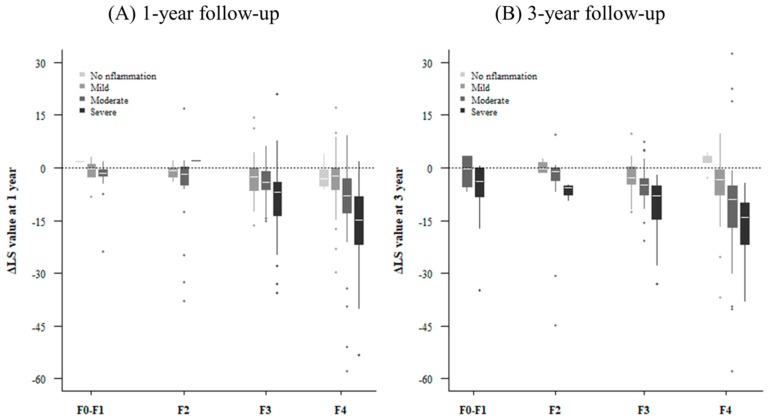
Change in liver stiffness value after (**A**) 1 year and (**B**) 3 years according to baseline fibrosis stage and inflammation grade.

**Figure 4 jcm-08-02065-f004:**
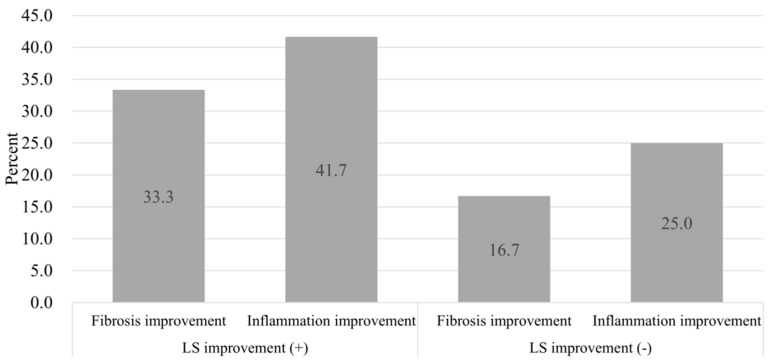
Change in liver stiffness value and histologic findings in patients who underwent repeated liver biopsies.

**Table 1 jcm-08-02065-t001:** Baseline characteristics of patients.

Variable	Baseline (*n* = 678)	1-year (*n* = 358)	3-year (*n* = 244)
Age, year, mean (SD)	47.12 (12.25)	48.54 (11.84)	49.57 (10.42)
Sex, n (%)			
Male	329 (48.5)	181 (50.6)	120 (49.2)
Female	349 (51.5)	177 (49.4)	124 (50.8)
Etiology, n (%)			
HBV	419 (61.8)	242 (67.6)	173 (70.9)
HCV	183 (27.0)	70 (19.6)	49 (20.1)
Alcoholic	12 (1.8)	7 (2.0)	5 (2.0)
Autoimmune	32 (4.7)	21 (5.9)	8 (3.3)
NAFLD	21 (3.1)	12 (3.4)	7 (2.9)
Others	11 (1.6)	6 (1.7)	2 (0.8)
BMI, kg/m^2^, mean (SD)	23.97 (3.44)	NA	NA
Laboratory findings			
AST, U/L, mean (IQR)	44.0 (30.0–77.0)	37.0 (22.0–38.0)	28.0 (17.0–30.0)
ALT, U/L, mean (IQR)	46.0 (29.0–83.0)	39.0 (26.0–39.0)	30.0 (17.0–30.0)
Total bilirubin, mg/dL, mean (SD)	1.11 (1.41)	0.82 (0.46)	1.02 (0.92)
Albumin, mg/dL, mean (SD)	4.15 (0.53)	4.35 (0.34)	4.34 (0.34)
Prothrombin time, INR, mean (SD)	1.01 (0.46)	0.97 (0.14)	0.89 (0.27)
LSM value, kPa, mean (IQR)	10.5 (7.3–19.6)	8.4 (5.3–12.0)	7.0 (1.8–10.0)
Liver biopsy, n (%)			
Fibrosis			
F0	13 (1.9)		
F1	96 (14.2)		
F2	132 (19.5)		
F3	186 (27.4)		
F4	251 (37.0)		
Steatosis			
No steatosis	370 (64.2)		
Mild	156 (27.1)		
Moderate	36 (6.3)		
Severe	14 (2.4)		
Inflammation			
No inflammation	28 (4.1)		
Mild	278 (41.0)		
Moderate	279 (41.2)		
Severe	93 (13.7)		

SD, standard deviation; HBV, hepatitis B virus; HCV, hepatitis C virus; NAFLD, nonalcoholic fatty liver disease; BMI, body mass index; AST, aspartate aminotransferase; IQR, interquartile range; ALT, alanine aminotransferase; INR, international normalized ratio; LSM, liver stiffness measurement.

**Table 2 jcm-08-02065-t002:** Linear regression analysis for the factors associated with baseline liver stiffness values.

Variable	Univariable		Multivariable	
	β (95% CI)	*p* Value	β (95% CI)	*p* Value
Age, year	0.175 (0.091 to 0.258)	<0.001		
Sex, male	0.158 (−1.857 to 2.174)	0.877		
Viral etiology	−5.530 (−8.487 to −2.573)	<0.001	−3.560 (−5.994 to −1.125)	0.004
BMI, kg/m^2^	0.115 (−0.223 to 0.452)	0.505		
Laboratory findings				
Platelet, 10^9^/L	−0.073 (−0.088 to −0.058)	<0.001	−0.024 (−0.039 to −0.010)	0001
AST, U/L	0.011 (0.005 to 0.016)	0.001		
ALT, U/L	0.005 (0.000 to 0.009)	0.032		
Total bilirubin, mg/mL	2.781 (2.153 to 3.409)	<0.001	1.904 (1.408 to 2.400)	<0.001
Albumin, mg/dL	−12.387 (−14.033 to −10.741)	<0.001	−5.120 (−6.862 to −3.378)	<0.001
Prothrombin time, INR	8.789 (6.859 to 10.719)	<0.001	4.913 (3.411 to 6.415)	<0.001
Creatinine, mg/dL	−0.706 (−1.775 to 0.363)	0.195		
Sodium, mEq/L	−1.428 (−1.809 to −1.047)	<0.001	−0.494 (−0.789 to −0.198)	0.001
Liver biopsy				
Fibrosis				
F0	1 (reference)		1 (reference)	
F1	−5.570 (−12.045 to 0.906)	0.092	−0.341 (−5.928 to 5.246)	0.905
F2	−3.765 (−10.153 to 2.622)	0.247	−1.240 (−6.819 to 4.340)	0.663
F3	1.072 (−5.176 to 7.319)	0.736	1.048 (−4.544 to 6.641)	0.713
F4	10.277 (4.048 to 16.506)	0.001	8.306 (2.730 to 13.882)	0.004
Steatosis				
No steatosis	1 (reference)			
Mild	0.868 (−1.562 to 3.299)	0.483		
Moderate	1.247 (−2.960 to 5.455)	0.561		
Severe	1.339 (−5.092 to 7.770)	0.683		
Inflammation				
No inflammation	1 (reference)		1 (reference)	
Mild	3.897 (−1.265 to 9.058)	0.139	2.347 (−1.487 to 6.182)	0.230
Moderate	7.414 (2.289 to 12.538)	0.005	3.374 (−0.489 to 7.237)	0.087
Severe	11.492 (5.977 to 17.007)	<0.001	6.476 (2.236 to 10.716)	0.003

BMI, body mass index; AST, aspartate aminotransferase; ALT, alanine aminotransferase; INR, international normalized ratio; CI, confidence interval.

**Table 3 jcm-08-02065-t003:** Linear regression analysis on 1-year liver stiffness value change amount.

Variable	Univariable		Multivariable	
	β (95% CI)	*p* Value	β (95% CI)	*p* Value
Age, year	−0.028 (−0.115, 0.060)	0.535		
Sex, male	0.263 (−1.785, 2.311)	0.801		
Viral etiology	3.527 (0.489, 6.566)	0.023		
BMI, kg/m^2^	0.086 (−0.274, 0.446)	0.638		
Laboratory findings				
Platelet, 10^9^/L	0.025 (0.008, 0.043)	0.005	0.027 (0.011, 0.042)	0.001
AST, U/L	−0.012 (−0.018, −0.007)	<0.001		
ALT, U/L	−0.011 (−0.017, −0.005)	<0.001		
Total bilirubin, mg/mL	−1.830 (−2.606, −1.055)	<0.001	−1.716 (−2.420, −1.011)	<0.001
Albumin, mg/dL	5.882 (3.793, 7.971)	<0.001		
Prothrombin time, INR	−5.399 (−6.994, −3.803)	<0.001	−4.647 (−6.126, −3.167)	<0.001
Creatinine, mg/dL	3.825 (−1.620, 9.271)	0.168		
Sodium, mEq/L	0.438 (0.051, 0.825)	0.027		
Liver biopsy				
Fibrosis				
F0	1 (reference)			
F1	3.909 (−4.872, 12.691)	0.382		
F2	1.493 (−6.893, 9.880)	0.726		
F3	−0.692 (−8.670, 7.285)	0.865		
F4	−2.352 (−10.337, 5.633)	0.563		
Steatosis				
No steatosis	1 (reference)			
Mild	−0.855 (−3.287, 1.576)	0.489		
Moderate	0.123 (−3.929, 4.174)	0.952		
Severe	−3.166 (−8.953, 2.620)	0.282		
Inflammation				
No inflammation	1 (reference)		1 (reference)	
Mild	−1.312 (−9.011, 6.387)	0.738	−1.256 (−8.298, 5.786)	0.726
Moderate	−3.925 (−11.578, 3.727)	0.314	−3.196 (−10.201, 3.808)	0.370
Severe	−9.875 (−17.673, −2.078)	0.013	−8.581 (−15.715, −1.447)	0.019

BMI, body mass index; AST, aspartate aminotransferase; ALT, alanine aminotransferase; OR, odds ratio; CI, confidence interval.

**Table 4 jcm-08-02065-t004:** Linear regression analysis on 3-year liver stiffness value change amount.

Variable	Univariable		Multivariable	
	β (95% CI)	*p* Value	β (95% CI)	*p* Value
Age, year	−0.013 (−0.137, 0.11)	0.832		
Sex, male	−2.451 (−5.004, 0.102)	0.060		
Viral etiology	−2.200 (−6.681, 2.281)	0.334		
BMI, kg/m^2^	0.042 (−0.372, 0.456)	0.842		
Laboratory findings				
Platelet, 10^9^/L	0.028 (0.005, 0.052)	0.020		
AST, U/L	−0.013 (−0.022, −0.005)	0.003		
ALT, U/L	−0.005 (−0.010, 0.001)	0.091		
Total bilirubin, mg/mL	−1.638 (−2.745, −0.531)	0.004	−1.129 (−2.153, −0.105)	0.031
Albumin, mg/dL	7.138 (4.336, 9.940)	<0.001	4.135 (1.187, 7.084)	0.006
Prothrombin time, INR	−7.089 (−10.301, −3.876)	<0.001	−5.954 (−8.978, −2.931)	<0.001
Creatinine, mg/dL	1.223 (−6.467, 8.912)	0.754		
Sodium, mEq/L	0.429 (−0.108, 0.967)	0.117		
Liver biopsy				
Fibrosis				
F0	1 (reference)			
F1	2.625 (−18.129, 23.379)	0.803		
F2	4.321 (−15.918, 24.561)	0.674		
F3	1.288 (−18.740, 21.317)	0.899		
F4	−0.766 (−20.821, 19.289)	0.94		
Steatosis				
No steatosis	1 (reference)			
Mild	−1.600 (−4.734, 1.534)	0.315		
Moderate	−1.515 (−6.960, 3.930)	0.584		
Severe	−6.865 (−13.645, −0.085)	0.047		
Inflammation				
No inflammation	1 (reference)		1 (reference)	
Mild	−5.772 (−14.596, 3.051)	0.199	−4.631 (−12.982, 3.721)	0.276
Moderate	−8.888 (−17.625, −0.151)	0.046	−6.553 (−14.959, 1.852)	0.126
Severe	−13.894 (−22.773, −5.015)	0.002	−10.725 (−19.299, −2.151)	0.014

BMI, body mass index; AST, aspartate aminotransferase; ALT, alanine aminotransferase; OR, odds ratio; CI, confidence interval.

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
