# Peer review of "The Influence of Histologic Inflammation on the Improvement of Liver Stiffness Values Over 1 and 3 Years"

_jcm, 2019, doi:10.3390/jcm8122065_

Round 1
Reviewer 1 Report
The authors addressed all of my concerns that were present in the original submission. That being said, in the new text the authors write, "First, it is well known that LS values can be overestimated by inflammation if the ALT level is higher than about 200 IU/L in current clinical practice" (lines 251-253). If this is well known, the authors should cite a study that supports this statement. Other than this, I have no concerns with this submission.
Author Response
Responses to the Associate Editor’s and Reviewers’ Comments
18 November, 2019
Dear reviewers and editorial staffs in Journal of Clinical Medicine
We are sincerely grateful for your thorough consideration and scrutiny of our manuscript, “The influence of histologic inflammation on the improvement of liver stiffness value during 1 year and 3 years”, control number jcm-649481. Through the accurate comments made by the reviewers, we better understand the critical issues in this paper. We have revised the manuscript according to the Reviewer’s suggestions. We hope that our revised manuscript will be considered and accepted for publication in the Journal of Clinical Medicine. We acknowledge that the scientific and clinical quality of our manuscript was improved by the scrutinizing efforts of the reviewers and editors.
The changes within the revised manuscript were highlighted (underlined and in blue). Point-by-point responses to the reviewers’ comments are provided below.
Reviewer #1 :
<GENERAL COMMENTS>
1) Reviewer’s comment: The authors addressed all of my concerns that were present in the original submission. That being said, in the new text the authors write, "First, it is well known that LS values can be overestimated by inflammation if the ALT level is higher than about 200 IU/L in current clinical practice" (lines 251-253). If this is well known, the authors should cite a study that supports this statement. Other than this, I have no concerns with this submission.
Author’s response: We appreciate the editor’s comment. As recommended by the reviewer, we added the reference of the sentence. The Discussion section of the manuscript was now revised as follows:
“First, it is well known that LS values can be overestimated by inflammation if the ALT level is higher than about 200 IU/L in current clinical practice.1-4” (lines 99-100)
References
Tapper EB, Cohen EB, Patel K, et al. Levels of alanine aminotransferase confound use of transient elastography to diagnose fibrosis in patients with chronic hepatitis C virus infection. Clin Gastroenterol Hepatol 2012;10:932-7 e1. Coco B, Oliveri F, Maina AM, et al. Transient elastography: a new surrogate marker of liver fibrosis influenced by major changes of transaminases. J Viral Hepat 2007;14:360-9. Arena U, Vizzutti F, Corti G, et al. Acute viral hepatitis increases liver stiffness values measured by transient elastography. Hepatology 2008;47:380-4. Sagir A, Erhardt A, Schmitt M, Haussinger D. Transient elastography is unreliable for detection of cirrhosis in patients with acute liver damage. Hepatology 2008;47:592-5.
Reviewer 2 Report
After the evaluation of the revised manuscript, i agree with this final version produced by the authors.
Author Response
Responses to the Associate Editor’s and Reviewers’ Comments
18 November, 2019
Dear reviewers and editorial staffs in Journal of Clinical Medicine
We are sincerely grateful for your thorough consideration and scrutiny of our manuscript, “The influence of histologic inflammation on the improvement of liver stiffness value during 1 year and 3 years”, control number jcm-649481. Through the accurate comments made by the reviewers, we better understand the critical issues in this paper. We have revised the manuscript according to the Reviewer’s suggestions. We hope that our revised manuscript will be considered and accepted for publication in the Journal of Clinical Medicine. We acknowledge that the scientific and clinical quality of our manuscript was improved by the scrutinizing efforts of the reviewers and editors.
The changes within the revised manuscript were highlighted (underlined and in blue). Point-by-point responses to the reviewers’ comments are provided below.
Reviewer #2 :
<GENERAL COMMENTS>
1) Reviewer’s comment: After the evaluation of the revised manuscript, i agree with this final version produced by the authors.
Author’s response: We deeply appreciate the reviewer’s kind comment.
This manuscript is a resubmission of an earlier submission. The following is a list of the peer review reports and author responses from that submission.
Round 1
Reviewer 1 Report
In this original manuscript, the authors describe the influence of histologic inflammation detected by biopsy on the liver stiffness level determined by transient elastography. ALT level is determined by a non-invasive method, however is not considered a reliable marker of hepatic inflammation, because can depends of several conditions, and may induce a false negative findings. Although biopsy is an invasive method, it can provide more precise information about of hepatic inflammation level and its location. By this way, I think the issue is up-to-date and relevant for research and clinical practice. However, I have some doubts about the additional and innovative information that this manuscript can provide about this issue, since there already are some published articles describing this effect. In the author’s opinion, which is the innovation of this original manuscript?
Comments:
The English has to be reviewed and improved. The authors use the expression “improvement of LS value”, when they really want to say that the LS value is increased. If there is an increase of LS value, there is not an improvement of LS. So, I think the authors should reconsider a better or/and more adequate expression to use. The table 1 should be improved. It is not easy to understand the table easily, because the contents are all listed without any criteria of importance. In line 200 and 201, if there is no statistical significance, the authors should not affirm that “improved LS group showed a high tendency to improve histological fibrosis and inflammation”. It seems a bias in the interpretation of the data. In line 219, the letter size is incorrect.Questions:
In the abstract, line 34 and 35, the authors say “Higher inflammation grade was a significant predictor for improvement of LS value…”. So, which is the consequence of this finding, for example, doctors should recommend a complementary biopsy (additional to ALT measurements) for some stage of hepatic inflammation? This project was submitted to an Ethic Committee, since it is being used data from the patients? Please include in the original manuscript, the name of Ethic Committee and reference code where the project was submitted and approved. Where are the data presented in the line 130 and 131? It is not possible to see these data in the figure 1. If there are more females than males in your study, why do you only present the nr of males? The authors could present both, but if you want to choose, it makes sense that you choose the gender in higher number in this population. In line 148, the authors say “…or other laboratory findings”. What findings? The authors must clarify that. In line 207 and 208, the authors say “…we found that LS value was affected even by low degree of inflammation”. However, the data only have statistical significance in the severe degree of inflammation. And there are other examples, where the authors have more optimistic considerations than they actually could have according their findings (see other example in line 264). Can you explain that? Like in the point 6, in line 254, the authors say “…showed that the degree of inflammation in addition to fibrosis had a clear influence on LS value”. Actually, fibrosis only presented statistical significance in the baseline for F4 degree. So, I think the authors cannot generalize by this way. Can you explain that? The same in line 256 and 257. If “…it was not statistically significant due to the small sample size…”, the authors cannot say that “the impact of inflammation change to LS value was confirmed.” Please, can you explain that?Reviewer 2 Report
The study by Yoo et al. describes the contribution of hepatic inflammation, fibrosis and other measures to liver stiffness at baseline, 1 year and 3 year time points. Out of the measures assessed, inflammation is the most consistent measure across time that is associated with liver stiffness and the improvement of liver stiffness. The strengths of this study are that it is a well-designed multi-center study and that multiple etiologies for chronic liver disease are investigated. That being said, there are some concerns I have with this manuscript as submitted:
One of the strengths of this study compared to others is including different etiologies in their patient cohort. That being said, this study is very similar to a study investigating liver stiffness and inflammation in hepatitis C patients (PMID: 25724309) as ~62% of the patients in this study had hepatitis C. Due to this, separating hepatitis C from hepatitis B (or combining other etiologies into groups) would add novelty to this report and should be performed if there are is enough statistical power to do so. At baseline, F4 fibrosis significantly correlates with liver stiffness, but this is not found at 1 year and 3 year follow up. The authors need to add text into the discussion on why this is occurring. A similar observation is found with steatosis where it does not influence baseline or 1 year, but severe steatosis does influence liver stiffness at 3 year follow up. The discussion should be expanded to discuss this in greater detail as well. The authors provide baseline characteristics (Table 1) for their patients. The authors should include this information for the patients at 1 year and 3 year follow up. Without this information, it is difficult to know the changes in the patients over time. For example, did BMI change over time in these patients? The authors need to expand the methods to describe how inflammation was defined into non, mild, moderate and severe. Supplemental figure 1 should be brought into the main manuscript text. There are wording/grammatical errors and the font changes on lines 218-219.